# Optimization of Sperm Management and Fertilization in Zebrafish (*Danio rerio* (Hamilton))

**DOI:** 10.3390/ani11061558

**Published:** 2021-05-27

**Authors:** Yu Cheng, Roman Franěk, Marek Rodina, Miaomiao Xin, Jacky Cosson, Songpei Zhang, Otomar Linhart

**Affiliations:** 1South Bohemian Research Center of Aquaculture and Biodiversity of Hydrocenoses, Research Institute of Fish Culture and Hydrobiology, Faculty of Fisheries and Protection of Waters, University of South Bohemia in Ceske Budejovice, Zatisi 728/II, 389 25 Vodnany, Czech Republic; ycheng@frov.jcu.cz (Y.C.); franek@frov.jcu.cz (R.F.); rodina@frov.jcu.cz (M.R.); xinmiao1206@126.com (M.X.); jacosson@gmail.com (J.C.); szhang@frov.jcu.cz (S.Z.); 2College of Life Science, Northwest University, Xi’an 710069, China

**Keywords:** zebrafish, *Danio rerio*, sperm motility, fertilization, short-term storage, extender

## Abstract

**Simple Summary:**

For scientific studies on the zebrafish model, simple and routine reproductive procedures should be used to ensure stable and repeatable results. When the milt is collected, spermatozoa are spontaneously activated by urine or excrement (low osmolarity) which routinely contaminates the samples, because of the minuscule size of the fish body. Therefore, whenever milt is collected from a zebrafish for short-term milt preservation and artificial fertilization, milt must be collected into an immobilizing solution, which because of its high osmolarity stops the movement of spermatozoa and keeps the sperm immobile until fertilization. Usually, the spermatozoa showed forward movement during the 35 s period following dilution in water. The sperm concentration ranged from 0.08 to 3.52 × 109/mL with a volume from 0.1 to 2.0 μL per male. The most suitable extender proved to be E400, which allowed storage of sperm for fertilization for 6 to 12 h at a temperature of 0–2 °C. To achieve a good level of fertilization and hatchability, a test tube with a precisely defined amount of sperm with extender, eggs and activating solution proved to be the most effective.

**Abstract:**

The aim of the present study was to investigate the spontaneous motility of spermatozoa and to optimize sperm collection, short-term sperm storage, and fertilization in zebrafish *Danio rerio*. The movement of spermatozoon in water was propagated along the flagellum at 16 s after sperm activation then damped from the end of the flagellum for 35 s and fully disappeared at 61 s after activation. For artificial fertilization, milt must be added to an immobilizing solution, which stops the movement of sperm and keeps the sperm motionless until fertilization. E400 and Kurokura as isotonic solutions were shown to be suitable extenders to store sperm for fertilization for 6 h. E400 stored sperm for 12 h at 0–2 °C. Sperm motility decreased only to 36% at 12 h post stripping for the E400 extender and to 19% for the Kurokura extender. To achieve an optimal level of fertilization and swim-up larvae rates, a test tube with a well-defined amount of 6,000,000 spermatozoa in E400 extender per 100 eggs and 100 µL of activation solution has proven to be more successful than using a Petri dish. The highest fertilization and swim-up larvae rates reached 80% and 40–60%, respectively, with milt stored for 1.5 h in the E400 extender at 0–2 °C.

## 1. Introduction

The zebrafish (*Danio rerio* (Hamilton)) is a popular freshwater fish belonging to the minnow family Cyprinidae of the order Cypriniformes, which has been extensively used as an invaluable vertebrate model organism for study in scientific laboratories since the late 1960s. Over the past few decades, laboratories across the world have produced numerous mutant strains and transgenic and wild-type zebrafish lines. As a model fish, it has the following advantages: easy and cheap maintenance in aquarium recirculation systems, short generation time, full genome map, year-round spawning, and rapid development. In particular, compared to larger fish species such as rainbow trout (*Oncorhynchus mykiss*) and common carp (*Cyprinus carpio*), it does not need breeding with hormonal induction [1,2]. In view of these advantages, it is appropriate to study gametes using zebrafish as a model organism for subsequent research, for example, into functional genomics [3], physiology [4], environmental monitoring [5], and human diseases [6]. It was found that the DNA methylome of sperm, which is crucial for embryonic development [7,8], was inherited in early zebrafish embryos. However, gamete handling in zebrafish as a model organism needs to be improved, as it provides an important basis for further research into epigenetics, in vitro fertilization, cryopreservation, artificial polyploidy, and uniparental inheritance induction.

It is known that the spermatozoa of most freshwater fishes are immotile in the testes and seminal plasma due to the level of osmolality and the composition of seminal plasma [9,10]. Activation occurs when spermatozoa are released into a hypotonic aqueous environment during spawning [9,10]. For zebrafish, sperm motility is initiated in hypotonic solutions at a wide range of osmolalities (0–270 mOsm/kg) [11,12,13] with the highest sperm motility at 150–210 mOsm/kg [13]. A few studies in freshwater species have shown that sperm can be contaminated with urine during collection by stripping. This results in spontaneous movement of spermatozoa in the collected milt of some cyprinid species such as common carp [14], tench (*Tinca tinca*) [15], asp (*Aspius aspius*) [16], and zebrafish [17,18]. In the case of large species of fishes, excrement or urine can be separated from the milt during collection [19]. In small fishes, these components cannot be separated when stripping through the genital papilla and must be collected (see Appendix A). Therefore, a higher incidence of milt contamination is possible in these small-bodied species. To prevent spermatozoa spontaneous movement due to low osmolarity of the urine or excrement, which results in a rapid decrease in fertility, it is important to collect milt in a suitable extender (immobilizing solution), which inhibits sperm movement and preserves its fertilizing ability [17,18,20,21]. After milt collection, the zebrafish laboratory guides recommend time limits to use gametes efficiently. These aging limits are 1.5 or 2 h for sperm stored in Hanks’ buffered salt solution on ice [22,23] and 2 h for eggs stored in a Hanks’ modified medium at room temperature [24]. Cardona-Costa et al. [25] indicated that the temporal limits usually recommended for zebrafish milt to fertilize fresh eggs could be extended for up to 24 h without significant differences, compared with fresh sperm in a modified medium (Hanks’ saline solution supplemented with 1.5 g BSA and 0.1 g ClNa, pH 7.4). There are several previously published extenders that show good short-time milt storage (from 6 to 24 h) such as Hanks’ balanced salt solution (HBSS) [26], modified Hank’s medium (MHBSS) [25], and the E400 extender [18].

Therefore, our goal was (1) to verify whether the phenomenon of spontaneous motility exists in zebrafish in general or only rarely, (2) to adjust the method of milt collection, (3) to test which extender is most suitable for milt management by the sperm motility parameters, fertilizing ability, and swim-up larvae rates, and (4) to optimize the fertilization methods.

## 2. Materials and Methods

### 2.1. Ethics Statement and Animals

The study was conducted at the Faculty of Fisheries and Protection of Waters (FFPW), University of South Bohemia in České Budějovice, Vodňany, Czech Republic. The facility has the competence to perform experiments on animals (Act no. 246/1992 Coll., ref. number 16OZ19179/2016-17214). The methodological protocol of the current study was approved by the expert committee of the Institutional Animal Care and Use Committee of the FFPW according to the law on the protection of animals against cruelty (reference number: MSMT-6406/119/2).

### 2.2. Fish Culture and Stripping

Zebrafish males (8–12 months old) were used and cultured in eight l aquaria with females separate. The male AB reference strain was used (wild type) from the European Zebrafish Resource Center, and the transgenic line vasa EGFP (basis from AB strain), expressing enhanced green fluorescent (EGFP) protein in germ cells, was exclusively purchased from the University of Liège, Belgium. An approximate sex ratio was 1:1, at a density of three fish per l at 28.5 °C in a recirculating water system. The photoperiod was set at 14 h light: 10 h dark, and fish were fed twice per day with Gemma micro 500 (Skretting, Norway) and once per day with baby brine shrimps (*Artemia* spp.).

Before milt collection, males were housed with females overnight in spawning aquaria as described in Franěk et al. [2]. In total, a set of 300 males and 10 females were anesthetized during these experiments, and among those, milt was collected from 200 males; the rest of the males ejaculated an insufficient volume of milt and were not included in the analysis. Males and females were anesthetized in TRIS buffered 0.05% tricaine methane sulfonate solution (MS-222; Sigma-Aldrich, St. Louis, MO, USA, E10521-10G) and briefly dipped in the aquarium water. The area of the fish’s genital papilla was gently dried with tissue paper, the fish was placed on tissue paper in dorsal recumbency (belly up), and milt was collected (Appendix A). The eggs were pooled during collection and fertilized promptly (Appendix A). Immediately after stripping and collection of gametes, fish were then transferred into fresh water for recovery and were not used any further in this study (Appendix A).

### 2.3. Experimental Design

The primary design of the experiment was as follows:(1)Testing spontaneous movement of spermatozoa.(2)Basic screening of four extenders from the literature with selections of the two most robust for further studies.(3)Detailed study of two extenders on how they are able to maintain motility and fertility depending on the storage period. Sperm motility, concentration, and seminal plasma osmolality were assayed using (a) milt from individual males, (b) pooled milt at time of collection, (c) individually collected milt and a storage pool of spermatozoa with good motility. In addition, visualization of the detailed movement of spermatozoa was performed using a high-speed video microscopy and a stroboscopic lamp. In the case of fertility, individually collected milt and a storage pool of spermatozoa with good motility were tested.(4)Finally, a suitable fertilization technique was specified.

### 2.4. First Step: Testing Spontaneous Movement of Spermatozoa

Testing whether spermatozoa are motile or not without activation by water is very important. Milt from five AB strain males was collected individually in a 10 µL micropipette using gentle, bilateral abdominal pressure. A drop of milt was immediately spread onto the surface of a glass slide previously positioned on a microscope stage at room temperature (22–23 °C) and observed at 20× magnification using dark-phase microscopy to evaluate whether global sperm movement occurred. Milt was not placed on ice (0–2 °C) during handling.

### 2.5. Second Step: Testing Extenders

Fifteen males from the AB strain were used to select the best extenders. Due to the spontaneous movement of spermatozoa in milt collected from all five males as observed in the previous test, improvement of the collection method was needed. The urogenital opening was carefully dried with tissue paper and then was rinsed with the extender to be tested. Milt was rapidly collected in a 10 µL micropipette using gentle, bilateral abdominal pressure and immediately added to 20 µL of the extender to be tested. Milt was collected separately from three males in each of 200 µL test tubes, each one containing 20 µL of the extender to assess the capacity of different extenders to preserve sperm motility. Milt collected individually from three males without extenders was used as a control group. Twelve males were used to test four extenders (prepared in the laboratory by the authors): (1) (Kurokura) (180 mM NaCl, 2.68 mM KCl, 1.36 mM CaCl_2_, and 2.38 mM NaHCO_3_) [15], (2) Hanks’ balanced salt solution (HBSS) (137 mM NaCl, 5.4 mM KCl, 1.3 mM CaCl_2_, 1.0 mM MgSO_4_, 0.25 mM Na_2_HPO_4_, 0.44 mM KH_2_PO_4_, 4.2 mM NaHCO_3_, and 5.55 mM glucose, pH 7.2, 300 mOsmol/kg) [26], (3) Modified medium (MHBSS) (137 mM NaCl, 5.4 mM KCl, 0.25 mM Na_2_HPO_4_, 0.44 mM KH_2_PO_4_, 1.3 mM CaCl_2_, 1.0 mM MgSO_4_, and 4.2 mM NaHCO_3_) (100 mL of Hanks’ saline supplemented 1.5 g BSA and 0.1 g ClNa, 320 mOsmol/kg, pH 7.4) [24,25], and (4) E400 extender (130 mM KCl, 50 mM NaCl, 2 mM CaCl_2_, 1 mM MgSO_4_, 10 mM D-(+)-Glucose, and 30 mM HEPES-KOH, pH 7.9, 400 mmol/kg) [18]. All extenders and samples of milt mixed with extenders were stored on ice; their spontaneous motility was estimated without activation by distilled water and then also activated in distilled water at 0.5, 6, 12, and 24 h post-stripping (HPS).

### 2.6. Third Step: Testing of Sperm Storage, Spermatozoa Concentration, Seminal Plasma Osmolality, and Fertilization with Swim-Up Larvae Rates

#### 2.6.1. Storage of Milt from Individual Males

Nine males including three males from the AB strain and six males from the EGFP strain were used for the milt storage experiment in extenders Kurokura and E400, which were selected as the best from the previous test. The urogenital opening of each male was dried with tissue paper and then its milt was collected directly into a 10 µL micropipette. After stripping the nine males individually, the milt from each male was immediately divided evenly into 200 µL tubes, one containing 20 µL of the Kurokura and the other with 20 µL of the E400. Altogether, there were 18 tubes with approximately 0.15–0.75 μL of milt in each tube; the ratio of milt: extender was 1:27–33 (see Appendix A). All samples of milt in extenders were stored on ice; the motility and velocity of the spermatozoa were evaluated at 0.5, 6, 12, and 24 HPS.

#### 2.6.2. Storage of Pooled Milt at Time of Collection

First, milt from one group of 11 AB strain males was one by one separated into one 200 μL tube with 80 μL of E400 and one 200 µL tube with 80 μL of Kurokura (altogether two tubes with approximately 5.5 μL of pooled of milt in each tube; the ratio of milt: extender was 1:15). Second, four groups of 5, 6, 11, and 12 AB strain males were similarly one by one evenly separated into four tubes with 39 μL of E400 and four tubes of 39 μL Kurokura, respectively. Altogether there were eight tubes with approximately 1.55–2.3 μL pooled of milt in each tube; the ratio of milt: extender was 1:17–25. All milt added into tubes with extenders was stored on ice, and the motility and velocity were evaluated at 0.5, 12, and 24 HPS.

#### 2.6.3. Individually Collected Milt and Storage Pool of Spermatozoa with Good Motility

Milt from nine individual AB strain males was evenly divided into 20 μL of E400 (9 tubes) and 20 μL of Kurokura (9 tubes). Then only sperm from six males with motility evaluated as good (above 90%) was pooled in two tubes with Kurokura or E400 extenders (c. 1.9 μL of pooled milt; the ratio of milt: extender was 1:63) and stored on ice. The motility and velocity were evaluated at 0.5, 12, and 24 HPS.

#### 2.6.4. Sperm Motility, Velocity, Concentration, and Osmolality of Seminal Plasma

Distilled water supplemented by 0.25% Pluronic F-127 (catalog number P2443, Sigma-Aldrich) used to prevent sperm from sticking to the slide was used as the activating medium (pH 7.0–7.5) and maintained on ice prior to all experiments. Sperm was activated at room temperature (21 °C) by mixing the diluted sperm sample (0.5–2 µL, according to the density of sperm sample) into 20 µL of the activation medium on a glass slide at 0, 6, 12, and 24 h post-stripping. The activated spermatozoa were directly recorded microscopically (UB 200i, PROISER, Madrid, Spain) at 20× using a dark-field condenser and an ISAS digital camera (PROISER, Spain) set at 25 frames/s. Analyses of the sperm recordings were performed by the Integrated System for Sperm Analysis software (PROISER, Spain) at 15 s post sperm activation, and 25 frames were used to analysis velocity parameters. Computer-assisted sperm analysis (CASA) included the percentage of motile sperm (%), curvilinear velocity (VCL, μm/s), and straight-line velocity (VSL, µm/s). The scale was calibrated. Escala X and Escala Y were both set up to 0.625 μm when using a 20× lens on a dark field microscope. Quantitative analyses of all samples were conducted in triplicate.

Sperm concentration together with the total number of sperm per male was evaluated for individual males in E400 and Kurokura extender solutions. In addition, spermatozoa of pooled sperm samples were counted. The sperm concentration (expressed as 10^9^/mL) was determined by a Bürker cell hemocytometer (Marienfeld, Lauda-Königshofen, Germany; 12 squares counted for each male) using an optical phase-contrast condenser and an ISAS digital camera (PROISER, Spain) under an Olympus microscope BX 41 (4009).

The osmolality of seminal plasma from 20 males was evaluated. A total of 15 μL of sperm was collected and centrifuged at 17,000× *g* at 4 °C for 5 min (Thermo Scientific, Fresco 21). The seminal plasma (supernatant) of 10 µL was collected and diluted with 50 µL (dilution six times) in distilled water. Finally, seminal plasma osmolality was repeatedly measured three times in a Freezing Point OSMOMAT 3000 (Gonotec, Berlin, Germany), and the mean result, multiplied by six, was expressed in mOsmol/kg.

#### 2.6.5. Evaluation of Sperm Motility by Stroboscopic Illumination

Milt was collected from three males and added to IS E400 (prepared in the laboratory by the authors). The following activation solution was composed of 1 part of IS E400 + 1 part 1% BSA + 5 parts of distilled water. For a detailed visualization of swimming spermatozoa, 0.5 µL of sperm with E400 was directly mixed with a 50 µL drop of activation solution, placed on a glass slide, previously settled on the microscope stage, and immediately after mixing, motility was video recorded under a final magnification of 200× or 400×. Motile spermatozoa were recorded within 10 s for visualization of all spermatozoa. The focal plane was always positioned in the vicinity of the glass slide surface. Video records were obtained with a S-VHS (SONY, SVO-9500 MDP) video recorder at 25 frames/s using a CCD video camera (SONY, SSC-DC50AP) mounted on a dark-field microscope (Olympus BX 50) with a stroboscopic lamp (Chadvick-Helmut, 9630, Ontario, CA, USA) and visualized on a video monitor. The stroboscopic flash illumination with adjustable frequency was set manually to 150–800 Hz depending on the time resolution needed. During the process of recording, the microscope stage was slowly manually moved back and forth: this allowed the visualization of multiple well-defined successive images of a moving sperm without overlap within every video frame [27].

#### 2.6.6. Visualization of Motility of Sperm Flagella by High-Speed Video-Microscopy

Methodology for visualization of fish sperm flagella motility parameters by high-speed video-microscopy was mostly according to Bondarenko et al. [28]. Briefly, in order to observe the detailed pattern of live fish sperm flagella, phase contrast optical microscopy with high magnification (40×–100×) objective lenses was used with oil immersion, resulting in a bright image of the very small diameter flagellum. The high-speed video recording provides high spatial and temporal resolutions (up to several 1000 images/s). Serial frames individually selected from such video recordings allow us to follow successive positions (every millisecond or less) of flagellum waves covering several full beat cycles. Such records allow the description of flagellar images during one or several beat cycles where several successive positions (up to 20) are available for detailed analysis. Analysis of each individual sperm cell image includes quantification of several flagellar parameters such as beat frequency (number of waves developed per second), amplitude and length of the successive waves, curvature of each wave, attenuation factor of waves along the flagellum, and curvature of the general wave pattern [29].

#### 2.6.7. Extender Evaluation by Fertilization, Swim-Up, and Malformation Larvae Rates

Milt from nine AB strain males was individually collected and evenly divided into 20 μL E400 and 20 μL Kurokura (a total of 18 tubes). Only sperm with high motility rate (>90%) in Kurokura and E400 extenders from six males were pooled and used for fertilization. Fertilization experiments were performed after 1.5 h of storage with pooled milt. Prior to fertilization, the concentration of the pooled spermatozoa in the two extenders was as follows: E400−0.063 × 10^9^ and Kurokura−0.059 × 10^9^ per mL.

Pooled eggs from five females (18 mg, about 80 eggs) were fertilized with 600,000 and 6,000,000 of spermatozoa and replicated four times. Total volume of milt with extenders and hatchery water was always maintained at 100 μL (Table 1). Extenders were added to the water from the hatchery to balance the ions to the level of the first group with a higher sperm concentration. Right after sperm addition, the tubes were shaken by hand for about 45 s. Finally, the fertilized eggs were gently distributed into small cell culture Petri dishes (3.5 cm diameter) filled with dechlorinated water and kept at 25 °C in an incubator (PolyLab, Poland). Each fertilization assay in each extender was replicated three times. The dead (white) eggs were removed at 1 h after fertilization, and the remaining eggs in each Petri dish counted as the initial total number of eggs. The eggs fertilized successfully were counted when embryos reached the 2–4 cells stage at 1.5–2 h after fertilization. Dechlorinated water was gently changed. The dead embryos were removed 24 h after fertilization and then the water was exchanged daily up to completion of swim-up larvae. The swim-up larvae (Figure 1A) and malformed larvae (Figure 1B; unusual body proportions, e.g., peritoneal or heart edema, irregular body axis, and head malformities) were manually counted on day 7 of incubation at 25 °C. For each Petri dish, the fertilization, swim-up larvae, and malformation rates were calculated as the ratio between the number of fertilized eggs or swim-up or malformed larvae relative to the initial number of eggs.

### 2.7. Fourth Step: Testing of Two Different Fertilization Methods

Milt from the AB strain was stripped as described previously and evenly divided into 20 μL of E400 extender and 20 μL of Kurokura extender; only sperm with high motility rate (>80%) from 10 males were pooled and used for fertilization. Fertilization experiments were performed with pooled milt stored on ice for 1.5 h. Prior to fertilization, the concentrations of pooled spermatozoa were 0.058 × 10^9^ in E400 extender and 0.067 × 10^9^ spermatozoa per mL in Kurokura extender. Sperm was used for fertilization at the same level as before of 6,000,000 spermatozoa per 100 eggs.

First, 1 mL cryotubes were used as small dishes for fertilization (see Appendix A). The eggs from six females (22.5 mg, about 100 eggs) were placed at the bottom of the tube and milt as drops added next to the eggs (not on them): (A) with 10 μL milt in E400 extender and activated with 90 μL hatchery water; (B) 9 μL milt in Kurokura extender and activated with 91 μL hatchery water (each time total volume = 100 µL). Second, small Petri dishes (35 mm in diameter and 15 mm in depth) were used for fertilization (see Appendix A). The same number of eggs was deposited to the edge of the Petri dishes by pipette tips and then fertilized directly with milt (it was necessary to keep the Petri dish tilted at a 45° angle): (A) 10.27 μL milt in E400 extender and activated with 400 μL of hatchery water; (B) 9.01 μL milt in Kurokura extender and activated with 400 μL hatchery water (total volume was c. 410 µL in each case). The difference between the two methods was that when using cryotubes, the number of spermatozoa in 1 µL of activation solution (water + milt with extender) was 60,000 spermatozoa, and in the case of Petri dishes, only 14,634 spermatozoa. Then the cryotubes and Petri dishes were shaken by hand for about 45 s. Finally, the fertilized eggs were gently distributed into large Petri dishes (9 cm diameter), filled with dechlorinated water, and incubated at 25 °C. Each procedure was replicated three times. Subsequently, incubation took place in a similar manner to the previous experiments.

### 2.8. Statistical Analysis

The data distribution homogeneity of dispersion was evaluated using Levene’s test. All the differences among means were determined by LSD test. The effect of extenders on motility (Figure 2A), VCL, and VSL (Figure 2B,C) at different storage times was performed by two-way ANOVA, respectively. The difference between AB and EGFP strain was tested by independent samples *t*-test, *p* < 0.05. Two-way ANOVA analyses were conducted to find the influence of storage time and individual males on sperm motility, VCL, and VSL (Figure 2 and Figure 3) and the effect of extenders with numbers of spermatozoa or fertilization methods on the fertilization and swim-up larvae rates of success (Figures 5 and 6). The results are presented as mean ± S.E. All analyses were performed at a significant level of 0.05 by using R [30].

## 3. Results

### 3.1. First Step: Testing “Spontaneous” Movement of Spermatozoa

A drop of milt was collected from a male and then, as soon as possible to avoid it drying up, which could occur within about 10 s, it was directly smeared on the surface of a glass slide so as to observe it under a dark-phase microscope. In all the samples collected from five males, motility of >70% of the spermatozoa was activated. The movement of spermatozoa was forwardly efficient at high speed (over 50 µm/s), not just local vibrations. Due to the high concentration of spermatozoa in a small volume (c. 0.1–1 µL), it was not possible to observe and record sperm movement for a prolonged period.

### 3.2. Second Step: Testing Extenders

To determine the best extender for zebrafish milt storage, milt from an individual male was collected and added into an extender. Right after mixing milt with an extender, sperm movement was immediately observed under the microscope. The same procedure was applied to the other three extenders. Three extenders, Kurokura, MHBSS, and E400, had a satisfactory sperm inhibition capacity, i.e., the sperm did not move or vibrate after dilution in any of the three extenders. On the other hand, few of the spermatozoa could be activated in HBSS. Sperm placed in Kurokura, MHBSS, and E400 solutions showed, after further activation by transfer in distilled water with storage of 6 h, the highest motility of around 43–65%. After 12 h storage in Kurokura, MHBSS, or E400, then transferred in water, the sperm motility of about 25–46% was recorded. However, there were only a few motile spermatozoa (<1%) observed in the HBSS extender at the same time (Table 2). In summary, it is concluded that Kurokura and E400 showed proportionally stable results after 6 and 12 h with slightly lower standard error than in the case of MHBSS. MHBSS deserves further experimentation, but due to the minimal volume of milt from individual males, it was necessary to choose only two extenders for further testing.

### 3.3. Third Step: Testing of Sperm Storage, Spermatozoa Concentration, Seminal Plasma Osmolality, and Fertilization with Swim-Up Larvae Rates

#### 3.3.1. Storage Sperm from Individual Males

The key quality parameter, motility rate of stored sperm, was measured in nine individual males. The influence of strains on motility parameters was not found. E400 and Kurokura were shown to be suitable extenders to store sperm for fertilization for 6 h at 0–2 °C. Significantly higher motility (16.6%) was observed with sperm stored in E400 than in Kurokura (*p* < 0.05) at 12 HPS. We observed similar differences regarding the VCL and VSL parameters. In total, there were no significant differences of sperm quality parameters (motility, VCL and VSL) between these two extenders at 0.5 and 24 HPS (Figure 2). On the other hand, the effective preservation of sperm movement was relatively low when after 6 h of storage, motility decreased by 35% compared to motility at 0.5 HPS. The difference after 12 h was 55% for the best extender, E400. After 24 h of storage, only 5% of the motile sperm was retained in the E400. The velocity of sperm movement, namely VCL and VSL at time 0.5 HPS, was at a level of 110 and 83 μm/s, and velocity gradually decreased at 24 h storage to 44 and 19 μm/s, respectively, in extender E400.

Two-way ANOVA analysis also showed that storage times and individual males had a significant influence on sperm motility, VCL and VSL, when sperm was stored in E400 and Kurokura extenders. In E400, for some males such as 1 and 4, there was a 10–20% motility rate after 24 h sperm storage (Appendix A) but the VCL in most of the males (except males 2 and 3) reached 50 µm/s (Appendix A). However, there was almost no motile sperm in Kurokura at 24 HPS (Appendix A) and the VCL was only observed in males 1 and 4 at 50 µm/s (Appendix A). Interestingly, after 0.5 HPS compared to 24 HPS, there was not much difference in VCL velocity. On the other hand, there was a rapid reduction in velocity within VSL. This means that within the VSL, there was a sharp drop in velocity at a straight track (Appendix A). Overall, the E400 proved to be of better efficiency to preserve zebrafish sperm motility when compared to the other extenders.

#### 3.3.2. Storage of Pooled Milt at Time of Collection

The motility of pooled sperm stored in E400 was significantly higher by 10% at 0.5 HPS than that stored in Kurokura. It decreased to <5% at 12 and 24 HPS in both extenders. There was no large variation of VCL between E400 and Kurokura at each storage time, but at 0.5 HPS, VSL decreased more in Kurokura than in E400 (Figure 3C). The VCL and VSL values continued to decrease, reaching 0 at 24 HPS (Figure 3C,E). Two-way ANOVA analysis showed that storage times and extenders had a significant influence on pooled sperm motility and VSL, but extenders had no effect on VCL (*p* < 0.05).

#### 3.3.3. Individually Collected Milt and Storage Pool of Spermatozoa with Good Motility

The motility of pooled good quality sperm in E400 and Kurokura extenders was similar at 0.5 HPS. Then it decreased to around 20% at 12 and 1% at 24 HPS in both extenders. There was no large difference of VCL and VSL values between E400 and Kurokura at each storage time (Figure 3B). The VCL and VSL continued to decrease to 40 and 20 μm/s, respectively, at 24 HPS (Figure 3D,F). Two-way ANOVA analysis showed that storage times but not extenders had a significant influence on sperm motility, VCL and VSL (*p* < 0.05).

The pooled results from individuals producing good quality sperm compared with milt directly pooled on collection showed that the former could be stored for a longer time in extenders. The motility was higher by 20% and VSL by 30 μm/s in pooled good quality sperm than in directly pooled milt during collection from males at 12 HPS. There were 1% motile pooled good spermatozoa with 40 μm/s VCL and 18 μm/s VSL and no motile spermatozoa in directly pooled milt during collection from males at 24 HPS.

#### 3.3.4. Sperm Concentration and Osmolality of Seminal Plasma

The range of spermatozoa concentration per male was 0.08–3.52 × 10^9^ per mL with volume from 0.1 to 2 µL. Sperm concentration had no significant influence on the sperm motility rates, but motility was different with storage time (*p* < 0.05). The range of total sperm quantity collected per male was 13,135–1,905,000 spermatozoa. Osmotic pressure values of diluted seminal plasma were measured as a mean of 44.67 mOsmol/kg, which corresponds to 268 mOsmol/kg as an initial osmolality of the seminal fluid.

#### 3.3.5. Visualization of Sperm Motility with Stroboscopic Light

When spermatozoa were transferred into distilled water, their flagellar motility was immediately activated (Figure 4A). Beating waves propagated along the flagellum usually with three crests of amplitude (one and a half sine wave lengths) at 16 s post sperm activation, as shown in Figure 4A; then, waves started to be slightly dampened in the distal portion of the flagellum, as illustrated at 35 s post sperm activation in Figure 4B. There was only a slight ripple (low amplitude wave) close to the head of the flagellum, indicating only vibration without efficient forward sperm movement at 61 s after sperm activation, as illustrated in Figure 4C. A cytoplasmic droplet was visible at the distal tip of the flagellum, indicating some damage to the flagellum due to the osmotic shock imposed by distilled water (Figure 4C,E). Figure 4D shows a real image of spermatozoa at 16 s post sperm activation without frame distribution at a light flicker frequency of 50 Hz. Finally, Figure 4E shows that at 1 min and 25 s post sperm activation, the spermatozoon was completely motionless.

#### 3.3.6. Visualization of Sperm Motility in High-Speed Video Images

Flagella motility was activated immediately after spermatozoa came into contact with water. Due to technical limitations, the earliest possible video record occurred at 4 to 5 s post mixing. During a time period from 5 to 15 s post-activation, three sine waves with about 5 µm amplitude and 9 µm wavelength were present along the flagellum; the beat frequency (number of waves generated per second) was 48+/− 4 Hz (Hertz or beat/sec) (see Appendix A, first part of the clip). Most spermatozoa describe circular tracks of c. 60 µm. As seen in the video images, flagellar waves have a three-dimensional shape, which imposed on the sperm cells a rotation around their progression axis. During a second period from 15 s to 1 min post-activation, sperm flagella continued to develop waves of similar amplitude but localized only in the proximal section closest to the head and absent in the distal flagellum. The flagellar beat frequency decreased to values ranging 15 to 25 Hz, and the sperm tracks became more linear, i.e., large circles (see Appendix A, second part of the clip). A third step in the motility period was reached starting from 1 min post-activation. Only a small fraction of the sperm population presented waves of low amplitude and low beat frequency proximally to the head, while the distal ¾ of the length of the flagellum was totally devoid of waves and straight. Some blebs can be seen along the flagellum as well as a tail tip curling, both resulting from damages of the flagellar membrane due to the very low osmolarity of water.

#### 3.3.7. Extenders Evaluation by Fertilization and Swim-Up Larvae Rates

The fertilization and swim-up larvae rates were slightly, but not significantly, higher when E400 was used compared to the Kurokura extender (Figure 5). The fertilization and swim-up larvae rates were significantly higher when 6,000,000 spermatozoa were used for fertilization (egg: spermatozoa ratio = 1:75,000) than 600,000 spermatozoa (egg: spermatozoa ratio = 1:7500). The highest fertilization and swim-up larvae rates reached 80% and 40%, respectively. Two-way ANOVA showed that the number of spermatozoa significantly influenced fertilization (*p* < 0.001) and swim-up larvae rates (*p* < 0.05), but different extenders had no significant effect on fertilization and swim-up larvae rates (*p* > 0.05). There were about 1–2% malformation differences between E400 and Kurokura, but the result was not significant.

### 3.4. Fourth Step, Testing of the Two Different Fertilization Methods

The results (see Figure 6) showed that a tube employed as a container for fertilization and swim-up larvae tests was significantly better than a Petri dish when using milt pre-incubated in E400. Two-way ANOVA showed that the fertilization method and/or spermatozoa concentration in activation water had a significant effect on the fertilization and swim-up larvae rates. Swim-up larvae rate using a test tube showed that E400 had a better swim-up larvae rate than Kurokura. There was about 0–2% malformations in each group without significant difference between E400 and Kurokura.

## 4. Discussion

### 4.1. Sperm Motility

The spermatozoa of zebrafish stored in the E400 extender remained immotile, but their motility was activated when transferred to a swimming medium such as fresh water (see Appendix A), where they presented a behavior similar to the spermatozoa other teleost species. After activation, they immediately acquired a rapid forward motility; then, after a period of 10 to 20 s, they slowed down gradually until the arrest of flagellum movement. Most sperm stopped moving after 60–90 s of activation. In other cyprinids such as common carp, it is also 60 s [31], and in tench, 40–60 s [15]. A decrease in swimming behavior in most other species could be primarily due to the decrease in energetic compounds (mostly ATP) during the motility period, where there is a trade-off between energy expenditure and motility duration [32]. It should be noted that when the head of the spermatozoon stopped movement, the flagellum continued to vibrate, although it had damaged membranes, especially at the tip.

Spermatozoa of freshwater fishes contaminated with urine showed a similar course of activation during stripping, leading to full arrest if an extender is not used to control the surrounding osmolarity so as to prevent motility. This category of fishes includes zebrafish [18,25,33,34] and the results of the present paper, European catfish (*Silurus glanis*) [21,35], tench [36,37], asp [16], tilapia (*Oreochromis mossambicus*) [38], and marine fishes such as Senegalese sole (*Solea senegalensis*) [39] and turbot (*Scophthalmus maximus*) [40]. In the case of zebrafish, this problem is even more critical because of the very small size of the fish, which makes it difficult to empty the urine from the bladder prior to milt collection (see Appendix A). In many fishes, urine accumulates in the bladder and is extruded together with milt during stripping [20,36,39]. It is known that spermatozoa in seminal plasma are immotile, but the hypo-osmotic level of urine changes the environmental conditions of the spermatozoa in the seminal fluid including osmolality, pH, and ion content. The last may spontaneously activate the spermatozoa [41,42]. This spontaneous activation of spermatozoa during the collection of milt destined for artificial fertilization is undesirable because the spermatozoa quickly lose their potency to fertilize [35,43]. All milt samples in our zebrafish study exhibited spontaneously active spermatozoa right after milt collection, because they were activated by contact with urine, evidenced by a mean osmolality value of 268 mOsmol/kg in the urine-contaminated seminal fluid. If milt was not contaminated by urine, the seminal fluid osmolality is predicted to be 288–315 mOsmol/kg, which is the osmolality of blood plasma in zebrafish [13,18]. It was shown that uncontaminated seminal plasma of many fish species presents an osmotic level similar to that of blood plasma [44,45,46]. Therefore, based on the current study, it is recommended that the urogenital papilla is washed with an extender prior to milt collection (see Appendix A), which prevents spermatozoa activation. The milt collected by such a procedure can then be stored in extenders.

Many studies have shown that when spermatozoa are activated by urine contamination, some energetic content is lost within a few seconds before being exposed to an immobilizing solution (IS) [43]. Before sperm is affected by IS, motility parameters change rapidly due to high post-activation energetic consumption. The relationship between motility, respiration, and ATP production was investigated in European catfish [47], common carp [48], and perch (*Perca fluviatilis*) [49]. Usually, half of the ATP was exhausted during the first 5–10 s after activation. This loss of energy can be restored by incubating sperm in IS or in artificial or natural seminal plasma [50,51]. Subsequently, the spermatozoa exhibit increased motility and fertilization capacity; this was demonstrated in common carp [50] and sterlet (*Acipenser ruthenus*) [51]. In summary, a longer storage time of sperm under the optimum osmotic, ionic, and pH conditions will maintain sperm quality and reconstitute energy.

### 4.2. Milt Storage

To prevent spontaneous movement and extend milt storage time, it was necessary to test the potentialities of extender solutions whose use has been previously published. From four commonly used extenders Kurokura, HBSS, MHBSS, and E400, it was found that the Hanks’ balanced salt solution (HBSS) did not stop sperm spontaneous movement. A few spermatozoa moved and vibrated (Table 2). It has been reported that the osmolality of HBSS and MHBSS was 300 and 320 mOsmol/kg [25,26]. We tested the HBSS osmolality and found it to be about 285 mOsmol/kg, which may be the reason the extender was unable to inhibit all movement. However, high osmolality was found in Kurokura, E400, and MHBSS of 370, 400, and 320 mOsmol/kg, respectively. Our results showed that after activation, there were >80% motile spermatozoa after 0.5 h of sperm storage and subsequently 43–65% motility after 6 h and even 25–46% motility after 12 h with sperm stored in Kurokura and E400. This proved that extenders with a higher osmotic pressure, >370 mOsmol/kg, can offset urine contamination and hypotonicity immediately after milt collection and keep sperm cells in an immobilized state. The sperm motility of common carp, another cyprinid, is inhibited by the high osmolality of the seminal plasma [10,52]; concentrations > 150 mM of KCl or NaCl have been found to inhibit their sperm movement [48,53]. The study by Wilson-Leedy et al. [33] provides crucial information about osmolarity control of zebrafish sperm motility. In this paper, an immobilizing solution was used (called ISS and composed of, in mM: 140 NaCl, 10 KCl, 2 CaCl_2_, 20 HEPES titrated to pH 8.5 with NaOH; final osmolality = 321 mmol/kg) and a range of osmolarities above and below this value were tested, which provides an important base for the choice of the correct osmolarity of the immobilizing solution. Some of our preliminary results (not detailed in the present publication) confirm the above-mentioned results of Wilson-Leedy et al. [33].

Zebrafish are characterized by a small body size, minute testes, and a minimal milt volume, i.e., <2 μL per male. These characteristics are similar to other small teleost fishes such as Japanese medaka (*Oryzias latipes*) [54], Monterrey platyfish *Xiphophorus couchianus* [55], and green swordtail *Xiphophorus hellerii* [56]. Furthermore, the sperm concentration between zebrafish individuals varied widely in the present study from 0.08 to 3.52 × 10^9^/mL, and the total spermatozoa number per male ranged from 13,135 to 1,905,000. Thus, it is necessary to pool milt from at least six males when it is intended to be used for artificial fertilization in the laboratory. Therefore, in the present study, a comparison of milt was made from (1) individual males saved individually, (2) individual collection and preservation of the best pooled sperm, and (3) direct collection and pooling. When the milt was collected from individuals and stored individually, spermatozoa motility decreased at 12 HPS and motility was significantly higher in E400 than Kurokura. There was no difference in VCL and VSL with the two extenders at 12 HPS, but later the VCL and VSL were a little higher in E400 than Kurokura. The preservation of milt did depend on the individual from which it was derived, and sperm from some males lost spermatozoa motility at 6 HPS. Milt individually collected and then pooled with only the best milt was essentially similar to milt stored individually. The worst results were obtained when the milt was mixed directly at collection time from males: the potential spermatozoa motility dropped at 12 HPS to 1–3% in this case. This could be expected, as we mixed good and bad milt, and bad milt probably affected negatively the overall ability to preserve spermatozoa.

It is well known that sperm plasma from bad sperm can significantly decrease the motility rate, VCL, and VSL of good sperm, even when the sperm is kept immotile in the bad seminal plasma [51]. There are many factors such as osmolality, Ca^2+^, Na^+^ ions, some enzymes, and proteins in maintaining sperm quality during storage [51]. Seminal plasma contains many nutrients, among which some are activating or inhibiting components. A similar phenomenon was found in stallions, where adding seminal plasma from low motility sperm (≤20%) to high motility sperm samples progressively reduced sperm motility [57]. As motility is the most important function of the spermatozoon, enabling it to reach the oocyte and fertilize it, simple mixing of milt with the extender E400 immediately at collection is recommended for aquaculture, but only if fertilization is performed relatively soon after, i.e., within about 30 min. In the case where it is necessary to store the milt for a longer period, then only individual male milt stored separately in the E400 solution, or pooled sperm with good motility stored in E400 solution, should be used within a time delay of 6 HPS or up to a maximum of 12 HPS.

### 4.3. Fertilization and Swim-Up Larvae Rates

In the present fertilization and swim-up larvae rates experiments, >6,000,000 spermatozoa for 80 to 100 eggs with 100 µL of activation solution (c. 90 μL of water + 10 μL milt and extender E400), i.e., a concentration of 60,000 spermatozoa per μL of activation solution, enabled 80% fertilization and 40–60% swim-up larvae rate to be achieved (see Figure 5 and Figure 6). Sperm concentration of 600,000 spermatozoa (egg: spermatozoa ratio = 1:6000) of 1.5 h storage was not successful for fertilization compared to 6,000,000 spermatozoa (egg: spermatozoa ratio = 1:60,000). However, Hagedorn and Carter [1] found that 4 × 10^4^ freshly collected spermatozoa was enough (70%) for the fertilization of 30 eggs (egg: spermatozoa ratio = 1:1333). The difference was probably due to the use of sperm aged 1.5 h in the present study. On the other hand, Hagendorn and Carter [1] did not state the level of hatching. Although in other fish species, such as European catfish, it was found to be better (increased sperm motility, VCL, and VSL when activated and fertilized with hatching level) to store milt for 1 day after stripping or 5 h before freezing in an immobilization solution [20,21,58]. In our study, zebrafish sperm aging is much faster than that in larger fresh- and warm-water fish species such as European catfish and common carp [59]. On the other hand, the preservation values of extenders achieved in zebrafish milt are close to those in another cyprinid, the tench, which also possesses milt easily contaminated with urine and must be collected in an immobilization solution [15].

A Petri dish with a larger volume of water for fertilization is routinely used for zebrafish during fertilization [2]. This traditional method was compared in this study to using a test tube with a volume of water that sufficiently covered 100 eggs. In both cases, the same number of eggs and the same number of spermatozoa were used. The ratio of spermatozoa: egg was also the same in both cases. It was found that there were better fertilization and swim-up larvae rates when using a test tube. This was due to (1) the shape of the containers; (2) the volume of water for fertilization (the test tube volume was four times lower); (3) E400 with milt being diluted 1:10 with water in the test tube and 1:40 in the Petri dish; (4) the osmotic concentration being higher in the test tube than in the Petri dish; and (5) there being a higher spermatozoa concentration in the test tube than in the Petri dish. In the test tube, the E400 extender provided the best hatchability. An optimum ratio of spermatozoa: egg is usually thought to be a key factor in in vitro fertilization [60], but the volume of activation water and number of eggs during fertilization are very important and must always be taken into account [61], as is also evident in the present study. Moreover, the test tube conditions also mimic more accurately the natural reproductive conditions that aquatic animals have adopted through long-term evolution processes [62].

## 5. Conclusions

In the current study, the movement of the zebrafish spermatozoon in water was ensured by waves (three curvatures) propagated along the flagellum at 16 s after activation. Later on, i.e., around 35 s post activation, the distal half of the flagellum appeared damped (no wave) and finally presented only some ineffective vibrations leading to full arrest. More than 70% of zebrafish sperm samples were activated spontaneously due to contamination by urine. Therefore, for the purposes of sperm preservation and artificial insemination, milt must be immediately mixed with an extender at collection, which fully stops the movement of spermatozoa. In addition, the area of the urogenital papilla must always be washed with an extender prior to sampling (see Appendix A). Without it, the fertility of zebrafish sperm cannot be preserved for even a few minutes. The extender E400 allowed storage of sperm for fertilization for 12 h at 0–2 °C. When milt was collected from individual males and stored individually, the sperm motility decreased to 36% at 12 HPS for E400 and to 19% for Kurokura extenders. The motility decreased by 35% at 6 HPS compared to 0.5 HPS. To achieve a sufficient level of fertilization and swim-up larvae rates, a test tube with 6,000,000 spermatozoa, 100 eggs, and 100 µL activation solution has proven to be better than using a Petri dish. Milt stored for 1.5 h in E400 extender at 0–2 °C resulted in the highest fertilization and swim-up larvae rates, 80% and 40–60%, respectively. Finally, it is recommended to use E400 or Kurokura solutions for the practical management of zebrafish.

## Figures and Tables

**Figure 1 animals-11-01558-f001:**
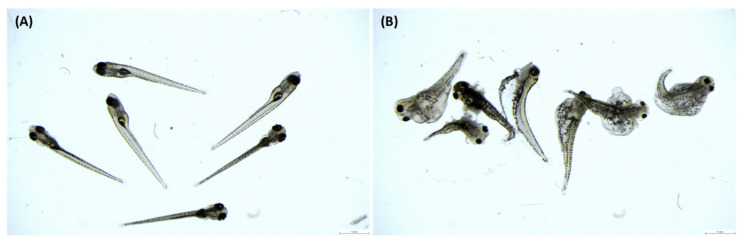
(**A**) Regular swim-up larvae and (**B**) malformed larvae of zebrafish at the age of 7 days (bar scale 1 mm).

**Figure 2 animals-11-01558-f002:**
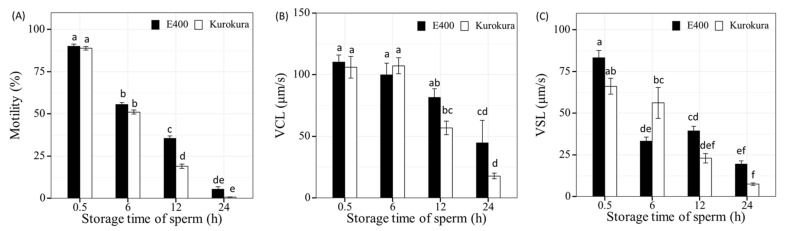
Motility parameters of zebrafish spermatozoa from all nine males at 15 s post-activation after milt storage of 0.5, 6, 12, and 24 h post-stripping (HPS) in Kurokura and E400 extenders: motility rate (%) (**A**); curvilinear velocity (VCL) (μm s^−1^) (**B**); and straight-line velocity (VSL) (μm s^−1^) (**C**). Mean ± S.E. with three replications are shown and compared by two-way ANOVA, followed by LSD tests. Groups with a common lower case letter do not differ significantly (*p* > 0.05).

**Figure 3 animals-11-01558-f003:**
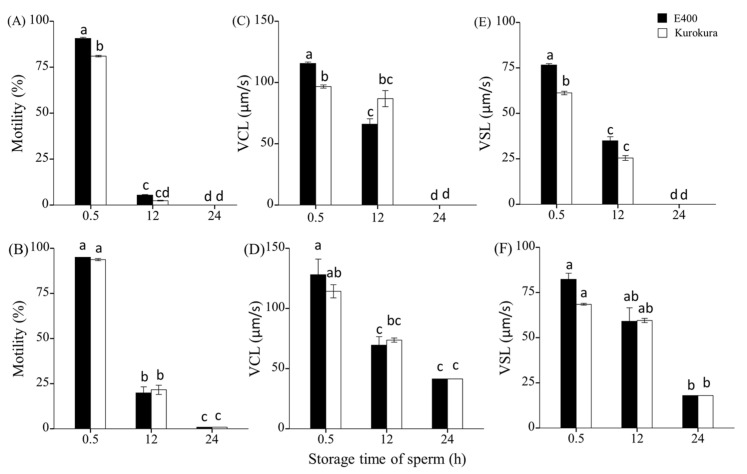
Results with storage of milt from directly pooled sperm (five pooled sperm groups and totally from 45 males) during collection from males (**A**,**C**,**E**) and with individual collection of milt and storage pool of good sperm (**B**,**D**,**F**): motility rate (%) (**A**,**B**); curvilinear velocity (VCL) (μm s^−1^) (**C**,**D**); and straight line velocity (VSL) (μm s^−1^) (**E**,**F**) at 15 s post-activation after milt storage of 0.5, 12, and 24 h post-stripping (HPS) in E400 and Kurokura. Mean ± S.E. with three replications are shown and compared using two-way ANOVA, followed by LSD tests. Groups with a common lower case letter do not differ significantly (*p* > 0.05).

**Figure 4 animals-11-01558-f004:**
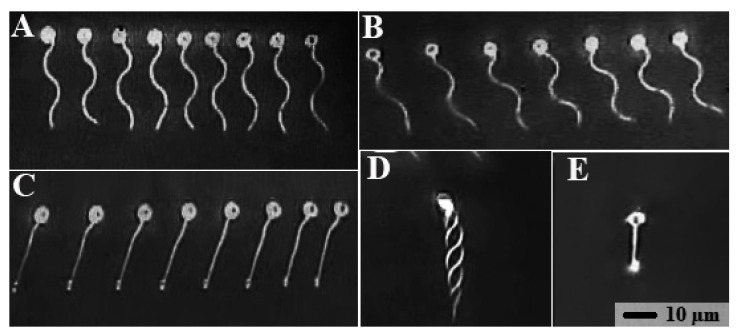
Images of actively swimming zebrafish spermatozoa observed by video microscopy under stroboscopic illumination. Motility was triggered by transfer of spermatozoa from seminal fluid and E400 into the activation solution. Dilution rate of milt with E400 was 1:100. Zebrafish sperm in panels (**A**–**E**), bar scale in (**E**): (**A**) swimming activation after 16 s; successive images of the same sperm cell illuminated by nine flashes per video frame; (**B**) as (**A**) but 35 s after activation showing dampening of waves in the flagellum (seven flashes per video frame); (**C**) as (**A**) but 61 s after activation shows slightly damaged flagellum (eight flashes per video frame); (**D**) as (**A**) but with overlapping images showing the flagellar envelope (flash frequency = 75 Hz); (**E**) 1 min 25 s after activation, any wave and consequently any swimming has fully ceased.

**Figure 5 animals-11-01558-f005:**
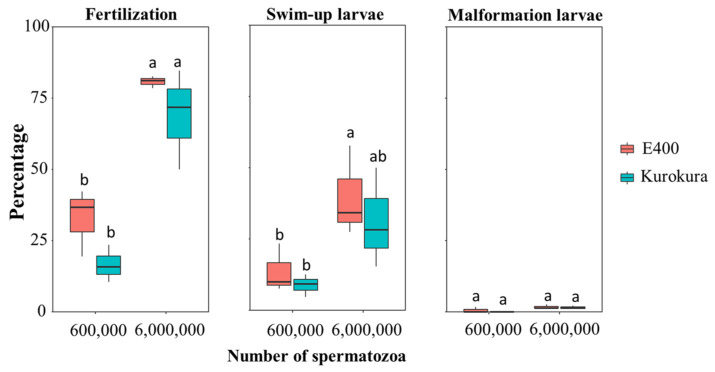
Evaluation of extenders by fertilization and swim-up and malformed larvae level using different amounts and storage times of milt for fertilization. Mean ± S.E. with three replications are shown and compared using two-way ANOVA, followed by LSD tests. Groups with a common lower case letter do not differ significantly (*p* > 0.05).

**Figure 6 animals-11-01558-f006:**
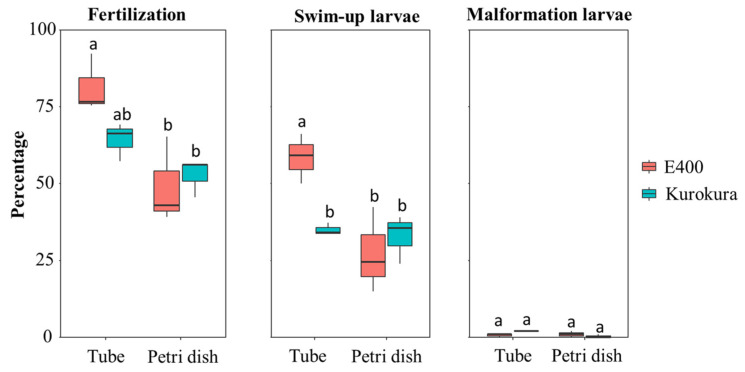
Testing of two test tube and Petri dish fertilization methods by fertilization and swim-up and malformed larvae level using extenders E400 and Kurokura. Mean ± S.E. with three replications are shown and compared using two-way ANOVA, followed by LSD tests. Groups with a common lower case letter do not differ significantly (*p* > 0.05).

**Table 1 animals-11-01558-t001:** Experiment design of fertilization with different pooled sperm numbers (six males) and extenders (E400 and Kurokura) at 1.5 h post stripping (HPS). Total volume of milt with extenders and hatchery water was 100 μL.

Sperm Number per 80 eggs	Extenders	Volume of Sperm without Additional Extenders (μL)	Volume of Additional Extenders (μL)	Volume of Sperm with Additional Extenders (µL)	Volume of Hatchery Water (μL)
600,000	E400	0.95	8.55	9.50	90.50
	Kurokura	1.00	9.00	10	90.00
6,000,000	E400	9.50	0	9.50	90.50
	Kurokura	10.10	0	10.10	89.90

**Table 2 animals-11-01558-t002:** Motility (%) of sperm stored in extenders (*n* = 3) and without extender (control, *n* = 3) activated by distilled water.

Extenders	Without Activation	Motility (%)			
		(Hours after Activation with Distilled Water)			
		0 h	6 h	12 h	24 h
Control	70% movement	0.0 ± 0.0 ^dA^	0.0 ± 0.0 ^bA^	0.0 ± 0.0 ^bA^	0.0 ± 0.0 ^bA^
Kurokura	No movement	80.0 ± 1.7 ^cA^	42.5 ±7.5 ^abB^	45.5 ± 3.5 ^aB^	1.2 ± 0.2 ^abC^
HBSS	<1% movement	87.4 ± 0.5 ^abA^	15.4 ± 1.7 ^bB^	2.7 ± 0.8 ^bC^	0.0 ± 0.0 ^bC^
MHBSS	No movement	89.0 ± 0.6 ^aA^	49.6 ± 7.0 ^aB^	30.3 ± 8.6 ^abBC^	6.8 ± 1.6 ^aC^
E400	No movement	82.5 ± 1.3 ^bcA^	65.0 ± 3.8 ^aB^	25.0 ± 5.0 ^abC^	0.0 ± 0.0 ^bD^

Without activation, observations were made directly after sperm collection (within 10–15 s) under a microscope without activation with distilled water. Data (mean ± S.E.) show the percentage of motile sperm. Different superscript lower case letters (a-d) indicate statistically significant differences between treatments at the same time point (0, 6, 12, and 24 h); different superscript upper case letters (A–D) indicate statistical differences between time points for each treatment (Control, Kurokura, …etc). A one-way ANOVA followed by an LSD test for post hoc multiple comparisons was used for analysis.

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
