# Peer review of "Optimization of Sperm Management and Fertilization in Zebrafish (Danio rerio (Hamilton))"

_animals, 2021, doi:10.3390/ani11061558_

Round 1
Reviewer 1 Report
Simple Summary: No comments
Abstract: In line 26, the sentence starts abrupt, authors can add little background, in 3-4 words, since this is abstract, there are word limits.
Introduction: In line 75 authors mentioned “the main zebrafish lab”, is there any reference for this lab?
Material and methods: In line 104, 8 1 aquaria with females……, the meaning is not clear.
Line 109, 0.25% Pluronic F-127, mention company name of this product.
Line 113, mention full form of HPS at least one time with abbreviation.
Line 136, IS E400, if this extender is bought commercially, mention company name.
Line 298, statistical analysis was performed in which software?
Results: Right down line 369, the figure top appeared. Need space. Also line 372, figure 3C, not aligned properly with 3A and 3B.
Discussion: No comment
Conclusion: It is not mentioned for future studies which extender is recommended clearly.
Reviewer 2 Report
Abstract
Please include in the abstract the aim of the study, any information about species, since there is no information throughout the abstract about species, which was used for the experiment.
Introduction
Please provide the better justification of the study, since extenders for storage of zebrafish sperm were tested before. What is new? Also the spontaneous motility in zebrafish was described by Matthews et al. 2018 (Changes to Extender, Cryoprotective Medium, and In Vitro Fertilization Improve Zebrafish Sperm Cryopreservation) Those authors indicated that that “urine contamination during collection of stripped zebrafish sperm reduces osmolality, causes premature activation, and is a source of variability in sperm quality. Therefore stripped sperm was collected from anesthetized males that had been thoroughly dried on a paper towel followed by drying of the urogenital opening with Kimwipe.”
I do not understand the aim of the study –Adjust the method of collection, since the Authors collected the semen as described by Matthews et al.(see above).
Methods
The description of the experiments is unclear. It is difficult to follow the author ( a lot of variants, pooling of semen, analysis using individual males).
To make it clear please provide the scheme of the entire experiment, including all steps (spontaneous movement of spz, testing extenders, sperm storage etc.
Why the Authors always have different ratio of milt : extender in the experiments? In my opinion it should be the same.
Statistical analysis
Regarding the data analysis, it is evident when reading the results that there was an interaction between time of storage (0.5; 6; 12; 24) and strain (AB and EGFP)(Fig. 3-4). However, the authors treat the results as if they had no interaction. It would be necessary for the authors to clearly describe the results they had and which had no interaction, presenting the P of this analysis. If it has no interaction, then it can analyze the variables with only the main effects. It is important to remember that in the descriptive part of the results, it is essential to start by saying whether or not the analyzed variable has an interaction, and only afterward if it is allowed to do the analysis and description of the main effects.
The statistical analyses must be reviewed according to what is found, the results must be modified accordingly.
In materials and methods Authors wrote “Two-way ANOVA analyses were conducted to find the influence of storage time and individual males on sperm motility, VCL and VSL (Figure 3, 4), however in the legend is one-way. I cannot find the difference on the Figures between different time of storage for each individual.
Results
Why for the further analysis the MHBSS was not used for further experiments. According to Table 1 , motility after 6 h and 12 h was comparable to E400 and Kurokura. After 6 h the motility was 49.6% (higher than in Kurokura) and after 12 h was 30.3 % (higher than in E400).
Line390 – it should be Figure 5C.
Line 391- it should be Figure 5C,E.
Figures –please provide the number of sampels used for the analysis. Figures 5, how many pools were used?
Point 3.4. “a tube employed as container for fertilization 478 tests was significantly better than a Petri dish when using milt pre-incubated in E400.” However for Kurokura there were no statistical difference.
Line 382- One way anova? In the legend two-way is indicate. It is not two-way Anova…
Discussion
What about discussion of the results of visualization of sperm motility. If the results are not discussed, so delete them.
Reviewer 3 Report
This manuscript concerns the collection of zebrafish milt and subsequent sperm storage. The authors discuss four choices of storage extenders and their suitability for milt management through assessment of sperm motility parameters at up to 24 hours post collection. Following this assessment, the authors discuss the application of the best two extenders to fertilization, comparing the use of Petri dishes vs test tubes. The work has been designed well with appropriate statistical tests. I am therefore recommending this work for publication subject to the authors addressing my minor comments, which are presented below.
Methods:
How many frames were used for sperm motility assessment with the Proiser system? The authors mention the capture frame rate but not the duration – this can have a significant impact on the parameters measured (particularly VCL).
Section 2.2.4.4:
This is well written but could do with a table to more clearly illustrate the process for each group.
Section 3.2:
What does ‘few sperm movement’ activated in HBSS mean – can this be quantified? If not, why not?
Table 1 is not clear. Is the ‘No activation’ column referring to the measurements before activation? Why does the control drop from 70% movement to 0% motility after activation with distilled water? The statistical significance indicators are not clear, what does each letter mean – letters are a to d but there are five rows in the table?
Section 3.3.1:
It is not clear what the authors mean by: ‘To preserve sperm a longer period of time was significantly higher (16.6%) stored sperm in E400 than in Kurokura (P<0.05) at 12 HPS.
Figure 3:
There appears to be little variation in VCL as storage time is increased, however the VSL appears to decrease by almost a factor of 4 over the same period. Can the authors comment on why the believe this is the case? Is there a qualitative change in swimming behaviour as storage time is increased?
There are a number of parts of this manuscript where the use of English could be improved to ensure that the author’s voice is clearly understood, particularly in the abstract and in the presentation of the results.
Round 2
Reviewer 2 Report
In my opinion the statistical analysis are still not correct.
Fig 2- “The effect of extenders on motility (Figure 2A), VCL and VSL (Figure 2B,C) at different storage times was performed by two and one-way ANOVA, respectively.
the results should be analysed using only two-way Anova. A two-way ANOVA tests the effect of two independent variables . In this case fist variable is the effect of extender and second the effect of time of storage. The significance of the two main effects (storage time and type of extender) and interaction should be marked on the Figures or Results section. Those results can not be analysed using one-way Anova. I do not understand why Authors analyses the results using one-way Anova . In the legend is also indicated that one-way Anova was applied.
Fig 3,-the same as above.
Fig. 5. Fig. 6- the results should be analysed using only two-way Anova :1) effect of extender, 2) effect of number of spermatozoa (Fig 5) or methods of fertilization (Fig 6). In the legend is also indicated that one-way Anova was applied.
The significance of the two main effects and interaction should be marked on the Figures or Results section.
Round 3
Reviewer 2 Report
I have no comments